# COMPOSITIONAL GENERALIZATION IN MULTIMODAL FOUNDATION MODELS

## ABSTRACT

The rise of large-scale multimodal models has paved the pathway for ground-breaking advances in generative modelling and reasoning, unlocking transformative applications in a variety of complex tasks. However, a pressing question that remains is their genuine capability for stronger forms of generalization, which has been largely underexplored in the multimodal setting. Our study aims to address this by examining sequential compositional generalization using COMPACT (Compositional Activities), a carefully constructed, perceptually grounded dataset set within a rich backdrop of egocentric kitchen activity videos. Each instance in our dataset is represented with a combination of raw video footage, naturally occurring sound, and crowd-sourced step-by-step descriptions. More importantly, our setup ensures that the individual concepts are consistently distributed across training and evaluation sets, while their compositions are novel in the evaluation set. We conduct a comprehensive assessment of several unimodal and multimodal models. Our findings reveal that bi-modal and tri-modal models exhibit a clear edge over their text-only counterparts. This highlights the importance of multimodality while charting a trajectory for future model development in this domain.

## 1 INTRODUCTION

Humans possess a remarkable ability to rapidly understand new concepts by leveraging and combining prior knowledge. This compositional generalization allows for an understanding of complex inputs as a function of their constituent parts. For instance, having grasped the meanings of "dax" and "walk twice" humans can effortlessly understand "dax twice" (Lake & Baroni, 2018). However, even as neural networks trained on increasingly larger datasets achieve impressive results across a wide range of tasks, their ability to compositionally generalize remains limited. Recently, the research community has demonstrated growing interest in evaluating models under different distributions, such as temporal shifts (Lazaridou et al., 2021; Liska et al., 2022), or unseen compositions (Lake & Baroni, 2018; Ettinger et al., 2018; Bahdanau et al., 2019; Surís et al., 2020). Within the domain of multimodal learning, prior investigations into compositionality have primarily delved into visual grounding (Thrush et al., 2022), downstream multimodal tasks like image captioning (Nikolaus et al., 2019; Jin et al., 2020) and visual question answering (Bahdanau et al., 2019), or vocabulary acquisition from videos (Surís et al., 2020) or with interactive agents (Hill et al., 2019).

Addressing the challenge of compositional generalization in the context of multimodal models becomes increasingly pertinent with the recent advancements in large multimodal foundation models, such as GPT-4 OpenAI (2023), Flamingo (Alayrac et al., 2022), and IDEFICS (Laurençon et al., 2023). This inspires us to investigate their potential for multimodal sequential compositional generalization, which we define as the model's capability to understand and generate predictions about novel compositions of primitive elements derived from sequential multimodal inputs – for instance, video data wherein actions unfold in a discernible order. Consider the process of cooking onions: one typically needs to *peel* and *slice* an ONION before *frying* it in a PAN. Our central inquiry revolves around the proficiency of models in comprehending such sequential and compositional activities.[1]

In this study, we introduce COMPACT (Compositional Activities) to investigate multimodal sequential compositional generalization, a uniquely constructed compositional dataset curated from the

---

[1]Note that this is different from in-context learning, where large-scale pretrained models are prompted to perform a task in a zero-shot setting, given a support set of task demonstrations.

| Inputs (key frames and utterances) | | | Targets (next utterance) |
|---|---|---|---|

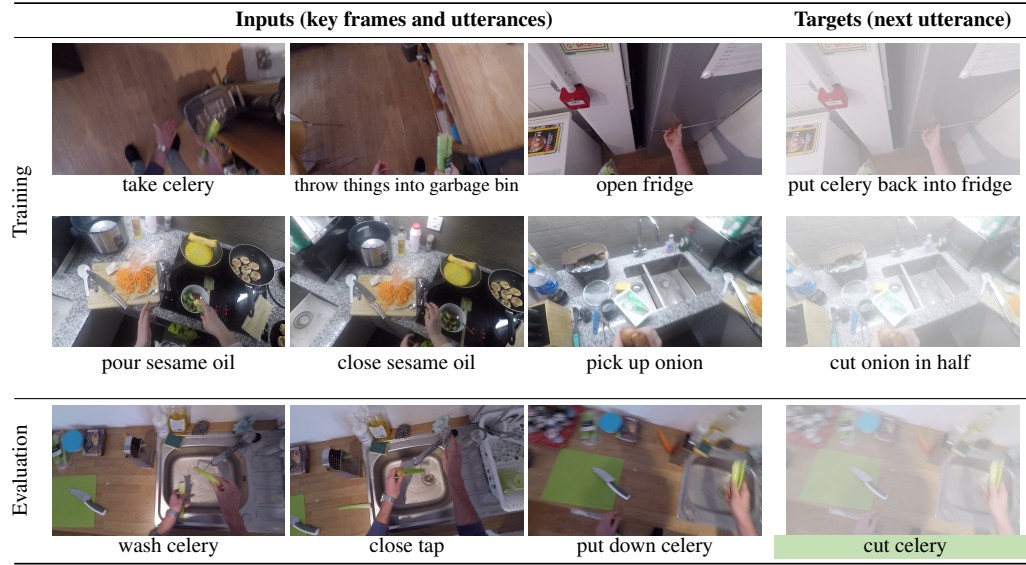

Figure 1: Overview of the compositional generalization setup in our COMPACT dataset. During training, the model has seen the verbs *wash*, *close*, *put down*, *throw*, *open*, *pour*, *cut* and *pick up* with the objects GARBAGE BIN, FRIDGE, SESAME OIL, ONION, and CELERY. It has never seen the composition of *cut* and CELERY, and thus needs to generalize to this novel composition at test time.

EPIC KITCHENS-100 dataset (Damen et al., 2022, EK-100). The EK-100 dataset encompasses 100 hours of egocentric video footage from 45 distinct kitchens, documenting individuals performing routine household tasks. Each video contains three streams of information: *visual data* in the videos; *audio data* involving non-narrative audio elements –such as the sounds associated with chopping an onion; and *textual data* in the form of short, crowd-sourced descriptions of the depicted activities, like "slice the carrot", "pick up the milk", or "wash the plate". From these descriptions, individual verb and object concepts such as *slice*, *pick up*, *wash*, and CARROT, MILK, PLATE can be extracted. The compositional splits are devised based on the verb and object concepts gleaned from the video descriptions, resulting in training and evaluation sets showcasing similar distributions of atomic concepts but featuring varied combinations therein. Consequently, models are necessitated to compositionally generalize from the training data. Aligning with the "dax twice" principle from Lake & Baroni (2018), if a model has been trained with videos illustrating how to *slice* various food items, excluding ROOT VEGETABLES, then it should be capable of compositionally generalizing to understand what it means to *slice* the ROOT VEGETABLES from previously unseen instances.

In our study, we conduct a comprehensive evaluation of publicly available models, encompassing encoder-only pretrained models such as ImageBind (Girdhar et al., 2023) and MERLOT Reserve (Zellers et al., 2022) in addition to (multimodal) large language models (LLMs) like LLaMA2 (Touvron et al., 2023) and IDEFICS (Laurençon et al., 2023). These models exhibit versatility in processing various combinations of input streams, ranging from language-only to combinations like video + language, video + audio, and even video + language + audio. Our key experimental finding indicates the formidable challenge that all of these models face in mastering compositional generalization. Yet, it becomes abundantly clear that the utilization of multimodal input sources yields discernible advantages, suggesting a promising direction for refining future models.

## 2 THE COMPACT DATASET

In our pursuit to systematically examine multimodal sequential compositional generalization, we devised the COMPACT dataset, leveraging sequences from the EK-100 dataset (Damen et al., 2022). As previously noted, each video in the EK-100 features first-person perspectives of unscripted kitchen activities occurring within natural household environments. A video is composed of a sequence of shorter clips, represented as $\mathbf{V} = (\mathbf{v}_1, \ldots, \mathbf{v}_k)$, each of which is accompanied by manually-annotated English narrations, denoted by $\mathbf{x}_1, \ldots, \mathbf{x}_k$, describing the activities within. Additionally,

these clips are integrated with audio tracks, $\mathbf{a}_1, \ldots, \mathbf{a}_k$, which contain the sounds of occurring actions. We define an *instance* in the dataset as a combination of video – audio – narration: $(\mathbf{V}, \mathbf{A}, \mathbf{X})$. Each instance consists of a window of 4 clips, with the initial 3 clips serving as context and the last one designated for prediction.

Given this dataset, our primary focus is to facilitate researchers in exploring how multimodal models compositionally generalize to unseen combinations of concepts. We meticulously curate the COM-PACT dataset to ensure a specific property: the individual concepts are consistently distributed across training and evaluation sets, while their compositions are novel in the evaluation set. This design mandates that a model should exhibit systematic generalization when interpreting the evaluation set. To illustrate, refer to the example shown in Figure 1. During training, the model comes across nouns such as CELERY, GARBAGE BIN, FRIDGE, ONION, and verbs including *take*, *wash*, *close*, *put down*. In our evaluation set, we specifically seek instances where an object-verb composition has not been previously encountered during training; for example, the unique pairing of the *cut* with the CELERY.

### 2.1 FORMING THE COMPOSITIONAL SPLITS

We use the *Maximum Compound Divergence* heuristic (Keysers et al., 2020) to curate a dataset that requires compositional generalization. The EK-100 dataset is annotated with 97 verb classes and 300 noun classes; these become the noun and verb *atoms*. Each instance in the dataset is assigned to the training / validation / test split based on the atomic and compound divergence (similarity) based on weighted distributions using Chernoff coefficient $C_\alpha(P\|Q) = \sum_k p_k^\alpha \, q_k^{1-\alpha} \in [0, 1]$ (Chung et al., 1989). To make atom distributions similar in train and test, we use $\alpha = 0.5$ for atom divergence. Here, we set $\alpha = 0.1$ to reflect that it is more important for a compound to be found in $P$ (train) rather than the probabilities in $P$ (train) and $Q$ (test) match exactly. Following this logic, we define compound divergence, and atom divergence for a train set $U$ and test set $W$ as follows:

$$\mathcal{D}_C(U\|W) = 1 - C_{0.1}(\mathcal{F}_C(U) \, \| \, \mathcal{F}_C(W))$$
$$\mathcal{D}_A(U\|W) = 1 - C_{0.5}(\mathcal{F}_A(U) \, \| \, \mathcal{F}_A(W))$$

where $\mathcal{F}_A(T)$ denotes frequency distribution of atoms, and $\mathcal{F}_C(T)$ denotes the distribution of compounds for a given set $T$ and $D_A$ and $D_C$ denote atom and compound divergences, respectively.

We calculated divergence scores for each instance until the atomic divergence of train and test set $D_A < 0.02$ and compound divergence of train and test set $D_C > 0.6$, which represents a sweet spot in terms of target distributions of atoms and compounds in the train and test sets (see Fig. 5 in the Appendix A.1). Finally, we randomly divide this test set into a validation and test set. The resulting dataset has 8,766 instances, which are split into 4,407 training, 2,184 validation, and 2,175 test instances. Please refer to Appendix A for the implementation details and Appendix B for a more detailed analysis of the COMPACT dataset.

## 3 MULTIMODAL SEQUENTIAL COMPOSITIONAL GENERALIZATION

Anticipating what comes next is a fundamental aspect of human cognition (Bar, 2007; Clark, 2015). From a cognitive perspective, it also serves as an engaging training paradigm (Baroni, 2020). In Multimodal Sequential Compositional Generalization, we seek to understand the extent to which multimodal foundation models are capable of understanding what comes next in activity sequences. We propose two tasks to measure multimodal sequential compositional generalization in the COM-PACT dataset: (i) next utterance prediction, and (ii) atom classification.

### 3.1 NEXT UTTERANCE PREDICTION TASK

The next utterance prediction task is a language generation problem, in which models need to predict the text narration that describes the final input in a sequence. Let $\mathcal{S} = (\mathbf{X}, \mathbf{V}, \mathbf{A})$ denote a triplet representing a short video clip with $\mathbf{X} = \{\mathbf{x}_i\}_{i=1}^K$ being a sequence of $K$ utterances, which describe a household activity and grounded with visual and audio signals, denoted by $\mathbf{V} = \{\mathbf{v}_i\}_{i=1}^K$ and $\mathbf{A} = \{\mathbf{a}_i\}_{i=1}^K$, respectively. This task involves generating the $(K + 1)^{th}$ utterance, $\mathbf{y} = \mathbf{x}_{K+1}$, following the preceding $K$ utterances and multimodal cues. The training data for this task consists of a set of sequences of microsegments, $\{(\mathcal{S}, \mathbf{y})\}$.

## 3.2 ATOM CLASSIFICATION TASK

The atom classification is a simplified form of the next utterance prediction task. Here, a model only needs to predict the verb and noun in the final input, in isolation of generating grammatically correct sentences. As such, it can be approached as a multi-class classification problem. Diverging from conventional action anticipation tasks (Damen et al., 2022; Gammulle et al., 2019; Ke et al., 2019), our unique setup allows us to approach atom classification through a compositional lens, enabling the prediction of verbs and nouns separately. More formally, let $\mathcal{S} = (\mathbf{X}, \mathbf{V}, \mathbf{A})$ denote a triplet representing a video clip with $\mathbf{X} = \{\mathbf{x}_i\}_{i=1}^{K}$ representing a sequence of $K$ utterances, which describe a household activity and grounded with visual and audio signals, denoted by $\mathbf{V} = \{\mathbf{v}_i\}_{i=1}^{K}$ and $\mathbf{A} = \{\mathbf{a}_i\}_{i=1}^{K}$, respectively. Our atom classification task involves predicting the verb/noun in the $(K+1)^{th}$ utterance, $\mathbf{y} = \mathbf{x}^{\mathbf{C}}_{K+1}$, following the preceding $K$ utterances and multimodal cues where $C$ denotes the verb or noun class.

# 4 MODELS

In our experiments, we benchmark a variety of neural network models on the proposed next utterance prediction and atom classification tasks, including several text-only (unimodal) and multimodal models to better understand the importance of different modalities in compositional generalization.

## 4.1 TEXT-ONLY UNIMODAL BASELINE (L)

Our first baseline is a text-only model to account for unexpected biases in COMPACT (Thomason et al., 2019). This is an encoder-decoder Transformer (Vaswani et al., 2017) with a hidden size of 256 units, where each microsegment is encoded within its own context. The model is trained using only the textual utterances $\mathbf{x}_{1:K}$ from the microsegment as the input, and the next utterance $\mathbf{x}_{K+1}$ as the target, *i.e.* to predict $p(\mathbf{x}_{K+1}|\mathbf{x}_{1:K})$. We use the same backbone in all multimodal baselines.

## 4.2 MULTIMODAL BASELINES

**Vision and Language (VL):** Our Vision and Language baseline encodes both textual and visual context for the next utterance prediction task. This model encodes the textual utterances $\mathbf{x}_{1:K}$ of each action from microsegments and the keyframe images $\mathbf{v}_{1:K}$ to predict the next utterance $\mathbf{x}_{K+1}$, *i.e.* $p(\mathbf{y} = \mathbf{x}_{K+1}|\mathbf{x}_{1:K}, \mathbf{v}_{1:K})$. This model is adapted from a model that parses a visual scene and learns cross-modal self-attention (Tsai et al., 2019) over textual inputs and visual data.

The visual inputs are encoded using pretrained CNN, and the textual inputs are encoded using a Transformer. More specifically, for the visual modality, we extract two types of features: one type represents global visual features, and the other represents object-level features. For the global features, we use a pretrained ResNet50 model (He et al., 2016) with ImageNet weights (Russakovsky et al., 2015). Object-level features are extracted using a Faster-RCNN object detector (Ren et al., 2017) with a ResNet-101 backbone (He et al., 2016) which is pretrained on MSCOCO (Lin et al., 2014) and finetuned on EK-100. We extract visual features from 5 objects for each keyframe. The resulting representation of a visual keyframe is the concatenation of the global and the object-level features. This concatenated vector is projected into a lower-dimensional space with a linear layer. The textual inputs are encoded using a Transformer with a 256D hidden layer.

The visual and textual modalities are then encoded by a cross-modal (CM) self-attention mechanism. In this model, we consider two modalities $\alpha$ and $\beta$, sequences of each modalities are denoted as $X_\alpha \in \mathbb{R}^{T_\alpha \times d_\alpha}$ and $X_\beta \in \mathbb{R}^{T_\beta \times d_\beta}$, respectively and $T_{(\cdot)}$ denotes sequence length and $d_{(\cdot)}$ denotes feature dimension. In this model, $\alpha$ is the language modality, and $\beta$ is the visual modality. In the cross-modal attention, the textual features are the *keys*, and the visual features are the *queries* and *values*, for aligning visual features to textual features. Let the Query be defined as $Q_\alpha = X_\alpha W_{Q_\alpha}$, the Keys as $K_\beta = X_\beta W_{K_\beta}$, and the Values as $V_\beta = X_\beta W_{V_\beta}$, where $W_{Q_\alpha} \in \mathbb{R}^{d_\alpha \times d_k}, W_{K_\beta} \in \mathbb{R}^{d_\beta \times d_k}$ and $W_{V_\beta} \in \mathbb{R}^{d_\beta \times d_v}$ are learnable weights. The cross-modal self-attention from $\beta$ to $\alpha$ is

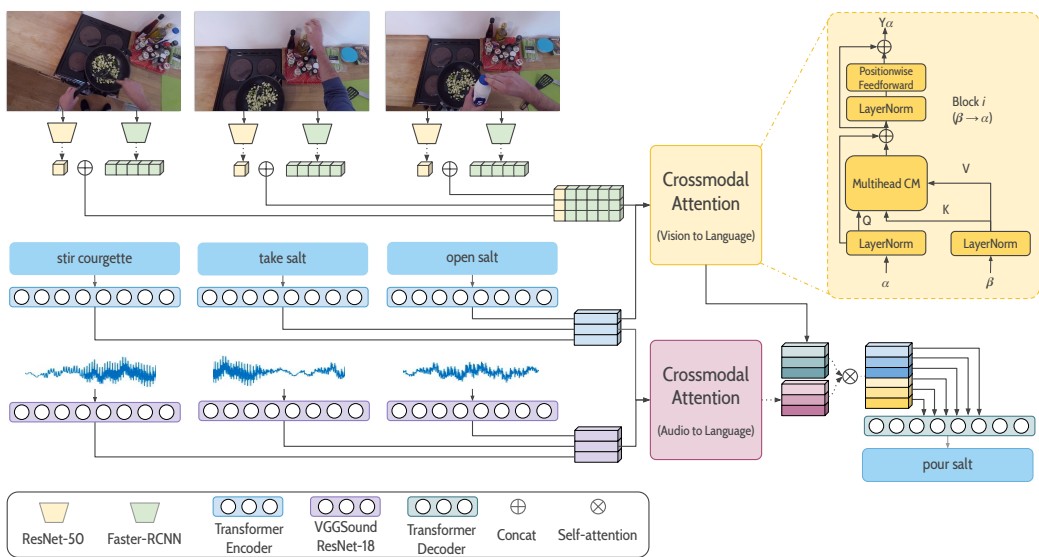

Figure 2: Overview of multimodal Audio, Vision and Language (AVL) baseline which integrates image, object-level, audio and textual features using two crossmodal self-attention blocks incorporated within a Transformer-based encoder and decoder architecture to predict the next utterance.

formulated as a latent adaptation $Y_\alpha \in \mathbb{R}^{T_\alpha \times d_v}$:

$$Y_\alpha = \text{CM}_{\beta \to \alpha}(X_\alpha, X_\beta) = \text{softmax}\left(\frac{Q_\alpha K_\beta^\top}{\sqrt{d_k}}\right) V_\beta \tag{1}$$

The output $Y_\alpha$ has the same length as $Q_\alpha$, but it is represented in the feature space of $V_\beta$. This enables the model to fuse different modalities, learning an alignment between the visual and textual features (see Eq.1). There are different strategies proposed in the literature for modeling cross-modal interactions and fusing different modalities (Xu et al., 2023). We fuse different modalities via a self-attention layer (Vaswani et al., 2017) over the aligned vision and language features, which are then fed to a 3-layer Transformer decoder with 4 attention heads that generates the next utterances.

**Audio and Language (AL):** The Audio and Language baseline has the same structure as the VL baseline. The key difference is that we represent the additional context using audio features instead of visual features. The model encodes both the textual utterances $\mathbf{x}_{1:K}$ and the accompanying audio data $\mathbf{a}_{1:K}$ to predict the next utterance $\mathbf{x}_{K+1}$, *i.e.* $p(\mathbf{x}_{K+1}|\mathbf{x}_{1:K}, \mathbf{a}_{1:K})$. The audio features are 512D vectors extracted using VGGSound (Chen et al., 2020), which is pretrained on 200K videos from YouTube videos totalling 550 hours of audio data. Here, the model learns a cross-modal attention over audio and textual features, analogously to the VL model, as inputs to a Transformer decoder.

**Object and Language (OL):** The Object and Language baseline once again uses the same architecture as VL baseline, but the visual context is represented using the labels of detected objects instead of continuous visual features. In this model, we embed object tags as a secondary set of textual features to our model along with the input utterances. Here, the object tags are represented as 292D one-hot encoded vectors (based on the number of unique tags) and projected to 256D with a simple linear layer. In this case, cross-modal attention aligns object tag features with language features.

**Audio, Vision and Language (AVL):** In the Audio, Vision, and Language (AVL) baseline, we leverage the audio, visual, and textual data using two cross-modal self-attention blocks. We use textual utterances $\mathbf{x}_{1:K}$ of each action along with the visual features $\mathbf{v}_{1:K}$ from the keyframes, and the VGGSound audio features $\mathbf{a}_{1:K}$ to predict the next utterance $\mathbf{x}_{K+1}$, *i.e.* $p(\mathbf{y} = \mathbf{x}_{K+1}|\mathbf{x}_{1:K}, \mathbf{v}_{1:K}, \mathbf{a}_{1:K})$. In this model, the input to the Transformer decoder is the concatenation of the audio-aligned textual features from the audio-textual cross-modal block with the visual-aligned textual features from the visual-textual cross-modal block (see Fig.2 for an overview).

**Object, Audio and Language (OAL):** The final baseline adds an extra modality to the OL baseline model to determine whether audio features affect the performance of a model that uses object tags. Here, we incorporate the extracted audio features from each microsegment to the OL model.

### 4.3 PRETRAINED MODELS

To comprehend the significance of large-scale pretraining, we conduct an extensive evaluation involving several publicly available models, namely LLaMA2 (Touvron et al., 2023), IDEFICS (Laurençon et al., 2023), MERLOT Reserve (Zellers et al., 2022, MerlotR), and Image-Bind (Girdhar et al., 2023). In our assessment, we investigate the performance of encoder-only models across both tasks, whereas auto-regressive models are evaluated exclusively through prompting within the context of the next utterance prediction task. It is worth noting that interpreting the performance of the pretrained models can be complicated as they may violate the distributional consistency between the train and test splits during their pretraining (Kim et al., 2022).

**Unimodal Models:** LLaMA2 is a large-scale text-only pretrained large language model trained on 500B tokens. We evaluate the LLaMA2-Chat 6.7B variant, as this version incorporates instruction tuning, resulting in much coherent and relevant predictions.

**Multimodal Models:** MerlotR learns to extract representations over video frames, text, and audio. The model is composed of an image encoder, an audio encoder, and a joint encoder that fuses textual, visual, and audio representations. This model employs contrastive span training, where an aligned span of audio and text is masked. In its training setup, the objective is to maximize representation similarity to an independent encoding of the masked audio and text spans. We extract multimodal audio and vision features through its pretrained encoder utilizing a similar backbone as in the VL model. ImageBind is a multimodal model that learns joint embeddings for 6 different modalities, including language, vision, and audio. It is trained only on image-paired data to bind the modalities together. We train a decoder using features extracted from the vision, language, and audio modalities. IDEFICS is a large-scale multimodal large language model based on Flamingo (Alayrac et al., 2022) architecture. It is composed of frozen language model and frozen vision encoder with learnable cross-attention blocks connection language and vision modalities. As Flamingo is not openly available and IDEFICS performs better than other open-source Flamingo implementations such as OpenFlamingo (Awadalla et al., 2023), we experiment with IDEFICS 9B version as the vision LLM. We prompt these models without any finetuning, and report 5-shot results for LLaMA2 and IDEFICS (see Section A.4 for prompting formats and Section E.3 for prompting ablations).

### 4.4 TASK-SPECIFIC CHANGES

For the atom classification task, we adapt the models described in the previous section by slightly modifying their architectures. In particular, we remove the decoder Transformer in these models and replace it with two fully connected layers, and train models by considering a classification objective that involves predicting either the verb or the noun in classifying the atoms.

## 5 EXPERIMENTAL SETUP

**Evaluation Metrics:** We use unigram BLEU (Papineni et al., 2002), Exact Match (EM), Categorical Accuracy (CA) and BERTScore (Zhang et al., 2019) metrics. The reported values represent the mean and standard deviation across 3 separate runs. In LLaMA2 and IDEFICS, we use nucleus sampling instead of separate runs. For EM, we calculate an accuracy score between the generated text sequence and the groundtruth. CA uses the verb and noun categories in EK-100 and calculates the categorization accuracy based on category match between the predicted sequence and groundtruth, *e.g.* the verbs *slice*, *dice*, and *chop* fall into the same verb category *cut*, and the nouns MOZZARELLA, PANEER and PARMESAN are grouped into the same noun category CHEESE. Hence, *slice* PANEER prediction is considered accurate if the groundtruth is *dice* PARMESAN.

**Training Procedure:** In next utterance prediction task, models are trained to minimize the negative log-likelihood of generating the next utterance, where the multimodal models are conditioned on additional modalities. Given the microsegment $\mathcal{S}$ and the model parameters $\theta$, the ob-

jective function is to minimize the negative log-likelihood of the $m$ tokens in the next utterance: $L(\theta) = -\sum_{i=1}^{m} \log p(y_i|\mathcal{S};\theta)$. In atom classification task, models are trained by attaching a MLP with a multi-class classification layer to the encoding of a microsegment $\mathcal{S}$. The objective function is to minimize the cross-entropy loss of predicting the expected atom (verb or noun): $L(\theta) = -\log p(\mathbf{x}_{K+1}^C|\mathcal{S};\theta)$. See Appendix D for implementation details.

## 6    RESULTS

### 6.1    ATOM CLASSIFICATION

Table 1: Quantitative comparison of baselines for Atom Classification. The best and the second best performing results are highlighted in bold and underlined, respectively.

| | Verb Classification | | | Noun Classification | | |
|---|---|---|---|---|---|---|
| | EM | CA | BERTScore | EM | CA | BERTScore |
| L | $13.37 \pm 0.5$ | $28.47 \pm 3.1$ | $75.16 \pm 0.6$ | $44.91 \pm 0.3$ | $51.83 \pm 0.3$ | $86.27 \pm 0.2$ |
| VL | $14.02 \pm 0.2$ | $28.68 \pm 2.3$ | $75.29 \pm 0.5$ | $42.72 \pm 0.7$ | $49.57 \pm 0.3$ | $85.81 \pm 0.2$ |
| AL | $13.76 \pm 0.3$ | $30.05 \pm 4.6$ | $76.26 \pm 0.5$ | $43.95 \pm 0.2$ | $51.11 \pm 0.5$ | $86.08 \pm 0.1$ |
| AVL | $14.06 \pm 0.7$ | $30.98 \pm 2.1$ | $76.12 \pm 0.7$ | $43.34 \pm 0.4$ | $50.43 \pm 0.9$ | $85.92 \pm 0.1$ |
| OL | $12.79 \pm 0.1$ | $29.97 \pm 1.5$ | $75.66 \pm 0.2$ | $44.35 \pm 0.9$ | $51.24 \pm 0.8$ | $86.00 \pm 0.3$ |
| OAL | $13.91 \pm 0.5$ | $29.90 \pm 1.3$ | $75.97 \pm 0.9$ | $43.83 \pm 0.9$ | $51.03 \pm 0.5$ | $85.92 \pm 0.2$ |
| MerlotR | $13.71 \pm 0.1$ | $\mathbf{33.50} \pm \mathbf{1.8}$ | $76.07 \pm 0.2$ | $\underline{45.42} \pm 0.6$ | $\underline{52.24} \pm 0.7$ | $\underline{86.42} \pm 0.1$ |
| ImageBind | $\mathbf{15.40} \pm \mathbf{0.2}$ | $31.54 \pm 2.5$ | $\mathbf{76.54} \pm \mathbf{0.4}$ | $33.67 \pm 0.3$ | $44.55 \pm 0.1$ | $83.96 \pm 0.1$ |
| MRH | 2.39 | 9.61 | 73.60 | $\mathbf{57.24}$ | $\mathbf{61.15}$ | $\mathbf{89.75}$ |

In Table 1, we present the outcomes of our atom classification task, which seeks to understand models' abilities to predict verb and noun atoms in isolation. For predicting verbs, we observe a similar trend in performance with the next utterance prediction task results. However, all models perform poorly in predicting nouns compared to the MRH baseline (Most Recent Heuristic). This baseline employs the most recently referenced object in the input microsegment as a prediction for the target noun, and most recently referenced verb as a prediction for the target verb. While the language-only baseline outperforms the multimodal baselines in noun prediction, we observe a slight improvement over the language-only model in predicting verbs within the multimodal models. This subtle yet noteworthy improvement underlines the value of leveraging multiple modalities for verb-related predictions.

### 6.2    NEXT UTTERANCE PREDICTION

Table 2: Next Utterance Prediction results on the test split. Using audio, visual, or object features consistently improves performance compared to the language-only unimodal baseline. The best-performing results are highlighted in bold, while the second-best results are underlined for clarity.

| Inputs | BLEU | EM | CA | BERTScore |
|---|---|---|---|---|
| L | $21.75 \pm 1.0$ | $2.89 \pm 0.3$ | $6.43 \pm 0.2$ | $79.06 \pm 0.1$ |
| VL | $31.25 \pm 0.3$ | $7.27 \pm 0.1$ | $12.95 \pm 0.4$ | $81.27 \pm 0.1$ |
| AL | $30.82 \pm 0.5$ | $6.81 \pm 0.5$ | $\underline{13.22} \pm 0.9$ | $81.20 \pm 0.0$ |
| AVL | $31.73 \pm 0.4$ | $7.04 \pm 0.4$ | $12.93 \pm 0.8$ | $81.50 \pm 0.1$ |
| OL | $30.79 \pm 0.6$ | $6.36 \pm 0.2$ | $12.21 \pm 0.1$ | $81.23 \pm 0.1$ |
| OAL | $\underline{32.02} \pm 0.2$ | $\underline{7.32} \pm 0.6$ | $13.08 \pm 0.9$ | $\underline{81.51} \pm 0.1$ |
| MerlotR | $31.50 \pm 0.3$ | $6.75 \pm 0.2$ | $12.85 \pm 0.1$ | $81.37 \pm 0.2$ |
| ImageBind | $\mathbf{33.52} \pm \mathbf{0.3}$ | $\mathbf{9.45} \pm \mathbf{0.5}$ | $\mathbf{15.04} \pm \mathbf{1.0}$ | $\mathbf{82.31} \pm \mathbf{0.2}$ |
| IDEFICS | $25.64 \pm 0.4$ | $5.76 \pm 0.1$ | $7.89 \pm 0.5$ | $80.92 \pm 0.1$ |
| LLaMA2 | $27.50 \pm 0.6$ | $5.36 \pm 0.6$ | $7.41 \pm 0.7$ | $78.76 \pm 0.2$ |

Table 2 presents the results of the next utterance prediction experiments. Notably, all multimodal models surpass the language-only baseline. Our baseline model that incorporates visual features (VL) exhibits consistent increases, showing gains of up to 9 BLEU, 4 EM, 6 CA, and 1 BERTScore points, compared to the language-only variant. Furthermore, harnessing a mix of audio, visual, and language features (AVL) or augmenting audio features with object tags (OAL) leads to additional improvements, highlighting the contribution of fusing multiple modalities. The most significant overall boost in performance is observed when visual features are utilized in the ImageBind pre-trained model, resulting in approximate increases of 11, 6, 8, and 2 points, for BLEU, EM, CA, and BERTScore metrics, respectively. The fact that LLaMA2 generates utterances with higher BLEU but lower BERTScore than IDEFICS suggests that LLaMA2 can better imitate the required dataset vocabulary than IDEFICS, even though IDEFICS produces more semantically plausible outputs. Conclusively, ImageBind's performance shows the advantages of employing pretrained multimodal features over merely merging separate unimodal encodings.

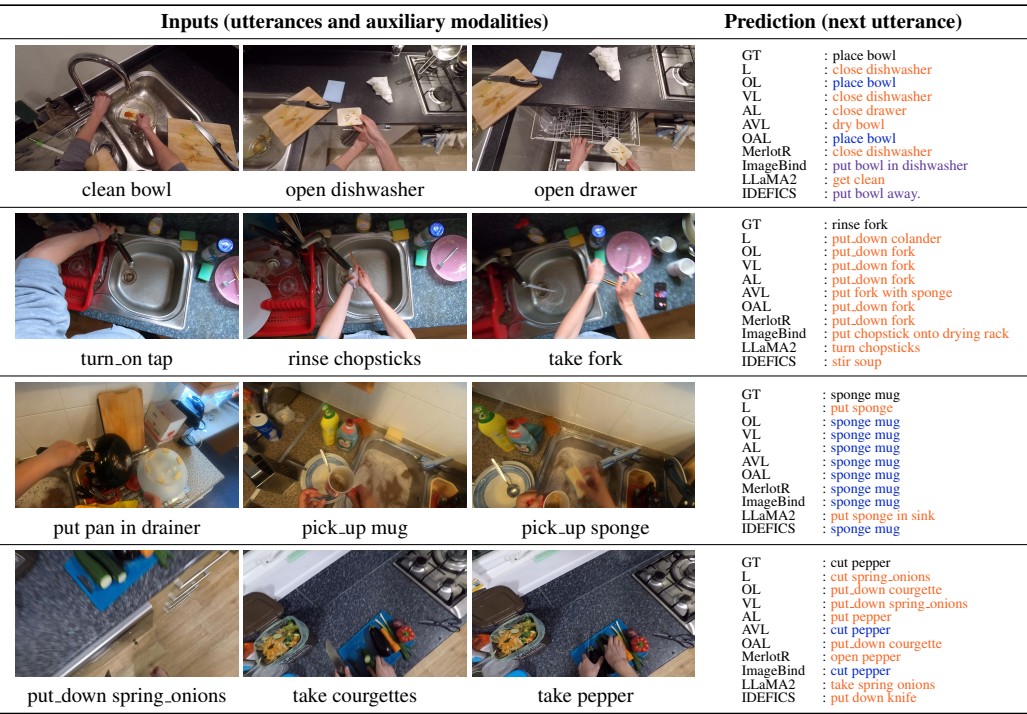

Figure 3: Next Utterance Prediction qualitative results. Models consider different combinations of input modality, as described in Section 4. Blue refers to correct, orange incorrect and purple semantically close.

In Fig. 3, we present a qualitative comparison of the baseline models via randomly selected examples from the test set. We believe that these illustrative examples effectively showcase the intricate and challenging nature of the proposed COMPACT dataset. During training, the models have never encountered compounds like *place* BOWL, or *rinse* FORK. In all of these illustrative examples, the text-only unimodal model fails to generalize to these novel compositions. However, in the first example, the OL and OAL baselines can predict the target composition correctly. ImageBind and IDEFICS, even though not exact matches, generates semantically plausible predictions. In the second example, all baselines struggle at predicting the verb while majority of the models predict the correct noun FORK, in line with performance discrepancy between verbs and nouns in Table 1. In the third example, all multimodal models can correctly predict the next utterance by leveraging the auxiliary modalities. Note that, LLaMa2 also fails in this example whereas IDEFICS can generate correct utterances. In the fourth example, AVL model and ImageBind models can correctly predict the *cut* PEPPER utterances. Interestingly, for this example both audio and vision inputs are needed, indicating that for sequential compositional generalization, models might have to leverage the available perceptual signal coming from different modalities at the same time.

## 7 RELATED WORK

**Compositionality.** Baroni (2020) studied the linguistic generalization capabilities of artificial neural networks. Lake et al. (2019) explored compositionality in a human-like few-shot setting, while others have studied compositionality at the representation level such as (Dasgupta et al., 2018; Ettinger et al., 2018). Unimodal compositional generalization datasets such as SCAN (Lake & Baroni, 2017), CFQ (Keysers et al., 2020), and COGS (Kim & Linzen, 2020) have been widely used in the literature to assess generalization abilities of neural networks. In parallel, researchers have been exploring different directions towards compositional generalization *e.g.* meta learning, (Lake, 2019), altering existing architectures (Akyurek & Andreas, 2021), and data augmentation (Qiu et al., 2022).

**Grounded Learning.** Johnson et al. (2017) studied systematic generalization in visual reasoning tasks. Bahdanau et al. (2019) investigated systematic generalization in a VQA-like context while Nikolaus et al. (2019) focused on compositionality to construct unseen combinations of concepts while describing images. Seo et al. (2020) transcribed speech to rank correct utterances in instructional videos. Surís et al. (2020) studied compositionality in word acquisition from narrated videos. Jin et al. (2020) investigated continual learning in unseen compound acquisition from paired image-caption streams. Other existing studies revolve around crafting conceptual benchmark datasets specifically designed to evaluate compositionality, *e.g.* (de Vries et al., 2019; Vani et al., 2021). Grounded compositional generalization is explored in (Ruis et al., 2020; Wu et al., 2021) within a 2D grid environment. Xu et al. (2021); Yun et al. (2023) investigated grounded compositional generalization for the concept learning problem. Li et al. (2022a) studied compositionality in a grounded setup with audio-language, Chen et al. (2021) leveraging audio-vision modality pairs.

**Foundation Models.** Recently, researchers have been studying foundation models to explore the possibilities of utilizing different modalities such as audio, vision, and text to solve grounded real-world problems (Guzhov et al., 2022; Girdhar et al., 2023; Driess et al., 2023). More recently, to assess visually-grounded compositional generalization capabilities of models, Bogin et al. (2021) proposed COVR, Zhuo et al. (2023) proposed ViLPAct, Ma et al. (2023) proposed CREPE. Unlike previous studies, our focus centers on a real-world audio, vision and language setting for compositional generalization (see Figure 1). We believe this study contributes towards better understanding the open challenges in multimodal sequential compositional generalization for foundation models.

## 8 CONCLUSION

**Limitations.** Despite the promising results, there exists a few limitations of our work. In our work, we introduce a novel dataset called COMPACT, that is carefully curated from the EK-100 dataset (Damen et al., 2022), which involves videos of daily kitchen activities, to dissect the impact of visual and auditory signals on linguistic compositionality. Hence, our conclusions may hinge upon certain domain-specific variables. It could be interesting to conduct future studies in an open-domain setting which might unravel additional insights. We investigate several different multimodal models for both next utterance prediction and atom classification tasks. However, it is important to note that for multimodal learning how to integrate different modalities is considered as an open research problem. In the literature, different strategies for multimodal data fusion have been proposed. Our experimental analysis could be further extended by considering some models that fuse the modalities in a way different than ours. More interestingly, from a systematic generalization point of view, an analysis could be carried out to explore the most effective fusion scheme. Finally, we acknowledge the textual utterances that we use in our work are inherently simplistic and do not capture all of the complexities in natural languages. Consequently, extending this work to a more natural source of language data that mirrors those complexities could be quite interesting direction.

**Conclusion.** In this paper, we present an investigation of linguistic compositionality and systematic generalization in a grounded setting for multimodal sequential compositional generalization. We show how a multimodal dataset can be utilized as a challenging test bed for this purpose. We design the next utterance prediction and atom classification tasks and follow a methodical approach in generating the training, validation and test sets for our compositional splits. We experiment with several baseline models and investigate models' ability to generalize to novel compositions and show how multimodal data can contribute towards solving systematic generalization problem and highlight major challenges. We hope our work will stimulate further research along these directions.

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

## APPENDIX

In the following, we provide a comprehensive set of supplementary notes that delve deeper into various aspects of our research:

- **Data Curation, Algorithms, and Preprocessing (Section A):** This section outlines the steps taken in data curation, algorithmic processes, and preprocessing techniques applied.

- **Exploratory Analysis of COMPACT (Section B):** Here, we present a detailed analysis of the COMPACT dataset, highlighting its unique characteristics.

- **Implementation Details and Reproducibility (Section D):** This section offers a detailed account of our implementation methodology, providing valuable information for those interested in replicating or extending our work.

- **Further Analysis (Section E):** We conduct additional analyses, expanding on key findings and offering deeper insights into the compositional generalization phenomenon.

- **Ethics Statement (Section F):** In this section, we present a comprehensive ethics statement detailing our commitment to ethical research practices throughout the study.

## A DATA CURATION, ALGORITHMS AND PREPROCESSING

### A.1 CURATING COMPACT: AN OVERVIEW

In our data curation and preprocessing for COMPACT, we leverage the EPIC-Kitchens-100 (EK-100) dataset, a collection of egocentric kitchen activity videos, which are split into shorter clips with accompanying narrations and audio tracks—referred to as "microsegments".

To curate our sequences of microsegments, we employ a window of 4 clips, with the initial 3 clips serving as context and the last one designated for prediction, yielding a total of 22,136 instances. We filter out repeated utterances that clearly represent a continuation of the same action, treating them as duplicates. Additionally, we exclusively consider text descriptions that share common nouns, ensuring that the noun mentioned in the target description also appears in the source text. This heuristic guarantees the presence of the target noun in both input sequences during both inference and training, allowing our setup to solely evaluate compositionality and systematic generalization.

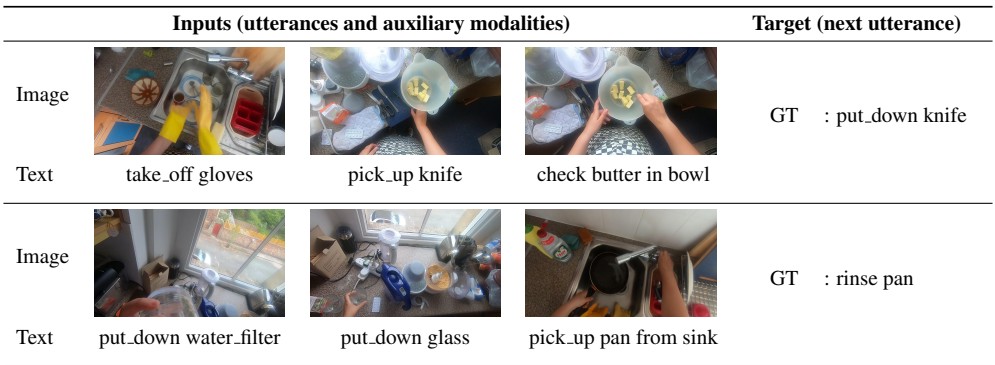

Figure 4: Curating dataset instances for compositional generalization. Targets such as *put_down* KNIFE and *rinse* PAN have never been observed by the learner during the training phase.

In our experimental setup, we introduce a scenario where a model must have prior exposure to all constituent atoms within a test instance, such as GRAB THE PLATE. WASH CUCUMBER. TAKE KNIFE., and is then tasked with predicting the subsequent utterance, such as SLICE CUCUMBER, during inference. It is important to emphasize that this target composition has never been encountered during the model's training phase (refer to Fig. 4). This setup allows testing models' ability to generalize to entirely unobserved compositions, even those with zero probability of occurrence in the training data. To create such dataset splits, we employ the Maximum Compound Divergence

(MCD) heuristic, crafting distributions that maintain similarity in the distribution of individual concepts (atoms), while deliberately introducing disparities in the distributions of concept combinations. In our case, we utilize 97 verb classes and 300 noun classes from the EK-100 dataset as the atoms. In particular, each sample is assigned to a specific split based on the atomic and compound divergence (similarity) based on weighted distributions using Chernoff coefficient (Chung et al., 1989). This process yields 8,766 instances, which are further partitioned into 4,407 for training, 2,184 for validation, and 2,175 for testing.

In Fig. 5, we visualize the atomic and the compound distributions over the constructed training, validation and test splits of our proposed compositional setup. Notably, these splits exhibit similar distributions concerning atoms while training and val/test splits do differ in terms of compounds.

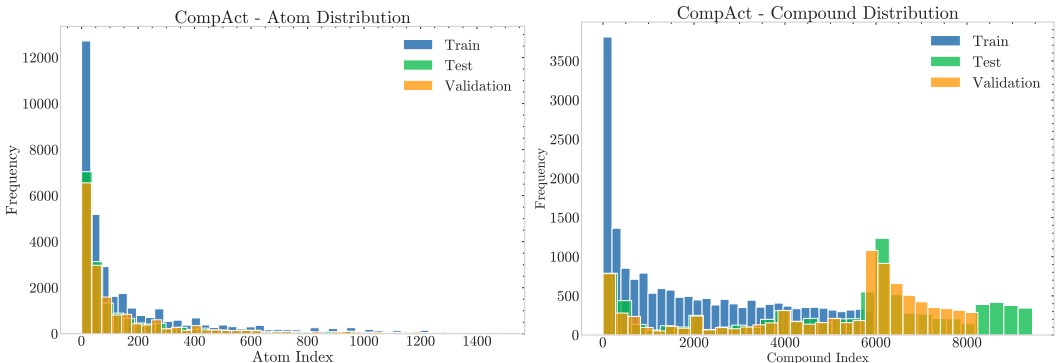

Figure 5: Plot on the left demonstrates the distribution of atoms while plot on the right shows the distribution of compounds for the train/validation/test splits in compositional split setup.

### A.2 ATOM AND COMPOUND SELECTION

In Algorithm 1, we describe the heuristic we use to create the compositional splits in COMPACT following the Maximum Compound Divergence (Keysers et al., 2020)

---

**Algorithm 1:** Split Generation Algorithm

---

**Data:** Dataset $M$
**Result:** Train split $U$, Test split $W$
1  Init $U, W$;
2  Init Atom Divergence $D_A$, Compound Divergence $D_C$;
3  Init $M_T$ with items in $M$;
4  Init $i$ to 0;
5  **while** $M_T$ *is not empty* **do**
6       Randomly choose $T \in \{U, W\}$ to add an item;
7       **if** $i = 0$ **then**
8          Randomly select and remove an item $m$ from $M_T$;
9          Add $m$ to split $T$ ;
10      **else**
11         Calculate $D_A$ for remaining items if added to $T$;
12         Filter items with $D_A$ below a threshold;
13         **if** *no items meet the criteria* **then**
14            Select item with highest $D_C$ as the best candidate;
15         **else**
16            Calculate $D_C$ for items if added to $T$;
17            Select the item with highest $D_C$ as the best candidate;
18         Add the best candidate item to split $T$ ;
19      Increment $i$ by 1;

---

### A.3 PREPROCESSING

### A.3.1 CHOOSING KEYFRAMES FROM VIDEOS

We adapt a straightforward yet effective approach to select representative images from each microsegment. We employ a simple heuristic to identify which keyframes to be selected for the span of the video clip. In particular, we run an object detector on the video frames and select the frames containing the highest count of object proposals detected by the object detector. This selection ensures that we capture the most visually informative frame from among the available candidates. In the case of ImageBind, we opt for the middle frame from each narration video.

### A.3.2 TOKENIZATION

As a preprocessing step, we replace multiword tokens with a single word. For instance, each occurrence of *put-down* is replaced with *put_down* and each occurance of OLIVE OIL is replaced with OLIVE_OIL. This preprocessing step is not applied to LLaMA2 and IDEFICS, since these models have their own vocabulary. Similarly, LLaMA2 and IDEFICS use their own tokenizers while other models simply use a whitespace tokenizer.

### A.4 PROMPTING FORMAT

In this section, we describe the heuristic we employ to formulate the inputs for our evaluation prompts targeting generative models. It is worth noting that the prompting templates for IDEFICS and LLaMA2, though similar, are not interchangeable as IDEFICS has the capacity to harness both visual and language data.

First, for both LLMs, we include an instruction at the start of the prompt as our language models are instruction-tuned. Then, we enumerate a set of few-shot examples. Finally, we provide the source section at the end of the prompt, leaving the target to be predicted.

### A.4.1 LLAMA2 PROMPT EXAMPLE

An example LLaMA2 5-shot prompt can be seen in the Fig. 6.

```
Predict the next narration given 3 sequential previous narrations from a cooking video
put down bowl . move frying pan . pick up spatula => put down spatula
put down bowl . move jar . pick up egg => crack egg
move yoghurt . put down bowl . pick up yogurt => put yoghurt
put down bowl . grab wok . move tap => lather wok
put down bowl . pick up spatula . stir meat pieces with spatula => put down spatula
pick up tins . put down tins . move bowl =>
```

Figure 6: Prompt template utilized for LLaMA2 evaluation.

### A.4.2 IDEFICS PROMPT EXAMPLE

An example IDEFICS 1-shot prompt can be seen in the Fig. 7. <Image n> denotes the image for the $n^{th}$ narration scene.

```
Predict the next action narration given 3 sequential previous actions (image-narration
pairs) in a cooking video.
put down bowl <Image 1> . move frying pan <Image 2> . pick up spatula <Image 3> =>
put down spatula
pick up tins <Image 1> . put down tins <Image 2> . move bowl <Image 3> =>
```

Figure 7: Prompt template utilized for IDEFICS evaluation.

# B    EXPLORATORY ANALYSIS OF COMPACT

In this section, we share an exploratory analysis of the COMPACT.

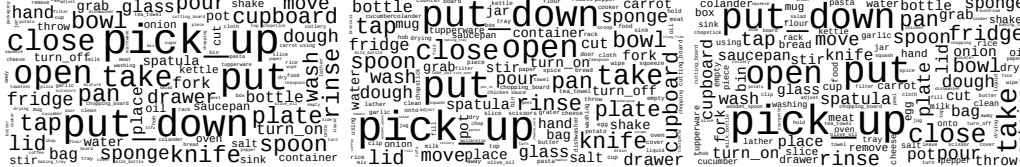

Figure 8: Word clouds for validation (left), test (middle) and train (right) splits.

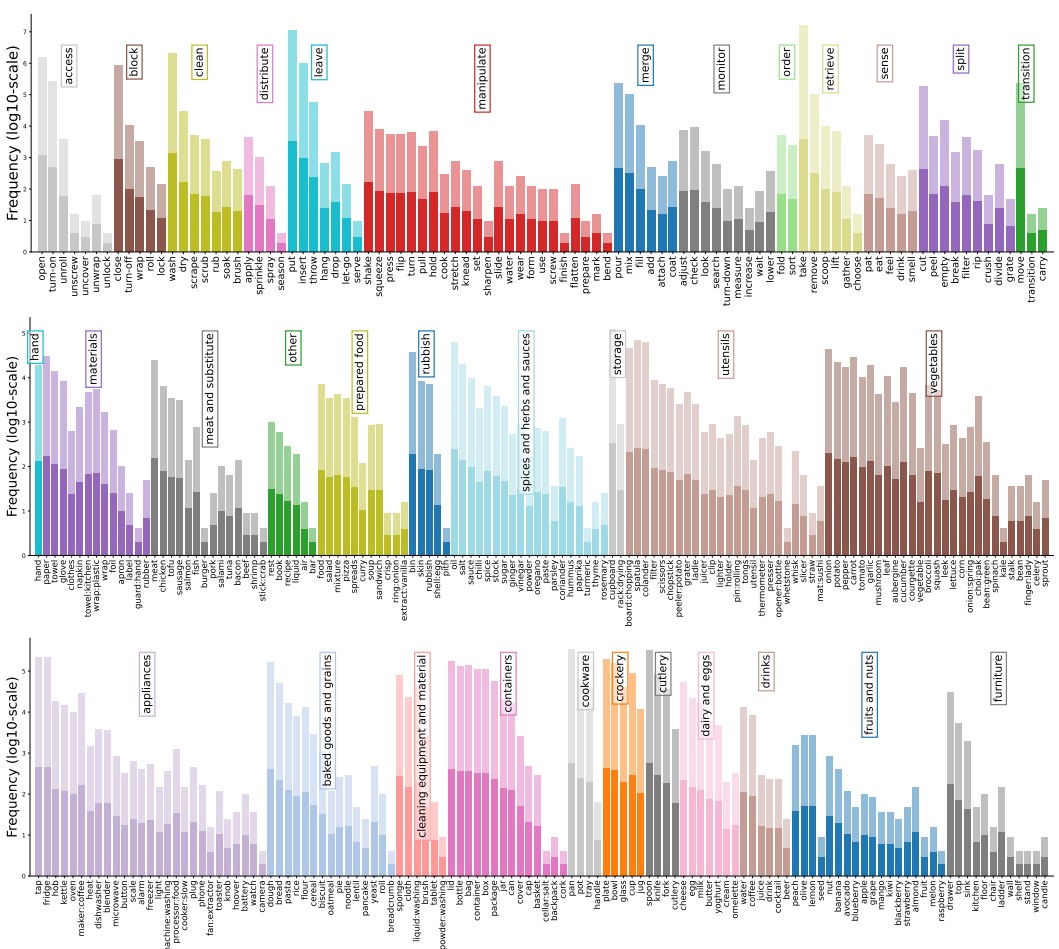

Figure 9: Distribution of verbs (top) and nouns (middle and bottom) from COMPACT

In Fig.8, we present the word cloud visualization for our compositionality splits consisting of all utterances. Fig. 9 illustrates the verb and noun distributions in the COMPACT dataset where validation and test splits are jointly stacked on top of the train split occurances and displayed in a lighter color.

# C    CHOICE OF EK-100 FOR COMPACT

The EPIC-Kitchens-100 (EK-100) dataset was chosen due to its established reputation in the research community and its densely annotated instructions, offering a rich and diverse dataset. It also has a clear segmentation of instructions, including verb and noun annotations, making it an

ideal candidate for curating the COMPACT dataset, allowing us to leverage audio, vision, and text modalities effectively.

Previously proposed datasets in the literature such as CrossTask (Zhukov et al., 2019) and GAIN (Li et al., 2022b) also consists of text instructions and multimodal components. Unlike CrossTask, which focuses on cross-task generalization, our study centers on compositional generalization. Similarly, while sharing similarities with GAIN in dataset formulation and the use of instructional videos, COMPACT differs in the description of atomic concepts and the mathematical definition of out-of-distribution (OOD) scenarios. We also conduct further analysis to evaluate whether the proposed benchmarks such as CrossTask or GAIN could be considered for a compositional generalization benchmark. Nevertheless, lack of proper annotations for atoms and compounds and the number of instances seems to be a challenge to generate compositional splits for these benchmarks.

## D   IMPLEMENTATION DETAILS AND REPRODUCIBILITY

For the reproducibility of our results, we plan to make the code, models, COMPACT splits and extracted features publicly available. All models are implemented with PyTorch.

### D.1   TRAINING REGIME AND HYPERPARAMETERS

We use the AdamW optimizer (Loshchilov & Hutter, 2017) with ReduceLROnPlateau learning rate scheduler to reduce the learning rate during training when validation BLEU plateaus. To train the models for next utterance prediction, we employ cross-entropy loss, initialize network weights via uniform distribution for both the encoder and the decoder. We use an early stopping strategy and stop the training if validation BLEU does not improve after a certain threshold. We clip gradients and set the gradient threshold to 0.1, and use a 3-layer multihead attention with 4-heads in the crossmodal self-attention block in all our multimodal models. We use the same strategy for atom classification, with one distinction where we use accuracy for early stopping and learning rate scheduler.

Table 3: Hyperparameters for each task. NUP refers to Next Utterance Prediction and AC refers to Atom Classification tasks.

| Task | Optimizer | LR | Batch Size | Patience | Scheduler Metric | Weight Decay | Dropout |
|------|-----------|-----|-----------|----------|-----------------|--------------|---------|
| NUP | AdamW | 3e-4 | 128 | 50 | BLEU | 5e-5 | 0.3 |
| AC | AdamW | 3e-4 | 128 | 50 | Accuracy | 5e-5 | 0.3 |

Hyperparameters for each task are given in Table 3. In our experiments, we use the ReduceLROnPlateau learning rate scheduler with the patience of 40. Following the insights from Csordás et al. (2021), we use the performance score as a monitoring metric for the scheduler (also early stopping) rather than using loss. For the NUP and AC tasks, we use BLEU and accuracy scores, respectively.

### D.2   MODEL SIZES AND TRAINING TIME

In Table 4, we present the number of trainable parameters and training time (MM:SS) for all of our trainable baseline models for both the next utterance prediction task and atom classification task.

LLaMA2 and IDEFICS experiments are ran on NVIDIA Tesla T4 and NVIDIA Tesla V100 GPUs respectively. Other experiments are run on NVIDIA 1080Ti GPUs.

Table 4: Model sizes and their training times for our experiments. Training times are averaged over 3 runs.

| | Next Utterance Prediction | | Atom Classification | | |
|---|---|---|---|---|---|
| Model | #params | Train Time | #params | Noun Train Time | Verb Train Time |
| L | 4.8M | 20:45 | 2.1M | 2:45 | 2:00 |
| OL | 12.0M | 38:15 | 9.3M | 12:15 | 8:30 |
| VL | 12.5M | 49:15 | 9.7M | 26:45 | 22:00 |
| AL | 12.0M | 40:00 | 9.3M | 10:45 | 8:15 |
| AVL | 12.6M | 52:00 | 9.9M | 15:30 | 13:00 |
| OAL | 12.1M | 39:00 | 9.4M | 14:15 | 10:30 |
| MerlotR | 12.1M | 18:30 | 9.4M | 6:30 | 4:45 |
| ImageBind | 8.4M | 28:15 | 5.7M | 9:45 | 8:30 |

# E   FURTHER ANALYSIS

## E.1   GENERALIZATION ON VALIDATION SPLIT

Table 5: Next utterance prediction results on validation split. Using audio, visual, or object features always improves performance compared to the language-only unimodal baseline. We report the mean and the standard deviation across three runs.

| Inputs | BLEU | EM | CA | BERT Score |
|---|---|---|---|---|
| L | $21.43 \pm 0.5$ | $2.88 \pm 0.1$ | $6.22 \pm 0.2$ | $79.2 \pm 0.1$ |
| VL | $30.59 \pm 0.4$ | $7.35 \pm 0.6$ | $12.39 \pm 1$ | $81.24 \pm 0.4$ |
| AL | $30.47 \pm 0.1$ | $7.06 \pm 0.3$ | $12.16 \pm 0.2$ | $81.19 \pm 0.1$ |
| AVL | $31.22 \pm 0.1$ | $7.44 \pm 0.4$ | $12.54 \pm 0.3$ | $81.44 \pm 0.1$ |
| OL | $30.50 \pm 0.4$ | $7.03 \pm 0.2$ | $12.42 \pm 0.8$ | $81.1 \pm 0.1$ |
| OAL | $\underline{31.42} \pm 0.1$ | $\underline{7.99} \pm 0.5$ | $\underline{13.36} \pm 0.2$ | $\underline{81.5} \pm 0.1$ |
| MerlotR | $31.36 \pm 0.4$ | $7.17 \pm 0.6$ | $12.68 \pm 0.5$ | $81.34 \pm 0.1$ |
| ImageBind | $\mathbf{34.13} \pm \mathbf{0.5}$ | $\mathbf{10.45} \pm \mathbf{0.8}$ | $\mathbf{16.08} \pm \mathbf{0.8}$ | $\mathbf{82.45} \pm \mathbf{0.2}$ |
| IDEFICS | $25.15 \pm 0.8$ | $5.66 \pm 0.5$ | $7.17 \pm 0.5$ | $80.75 \pm 0.2$ |
| LLaMA2 | $26.52 \pm 0.5$ | $5.37 \pm 0.3$ | $6.99 \pm 0.4$ | $78.59 \pm 0.1$ |

Table 6: Atom classification results on validation split. We report mean across three runs. Best and second best performing results are highlighted in bold and underlined, respectively.

| | Verb Classification | | | Noun Classification | | |
|---|---|---|---|---|---|---|
| | EM | CA | BERT Score | EM | CA | BERT Score |
| L | $12.92 \pm 0.8$ | $28.96 \pm 3.3$ | $74.96 \pm 0.4$ | $\underline{44.78} \pm 0.8$ | $\underline{52.28} \pm 0.8$ | $\underline{86.35} \pm 0.1$ |
| VL | $14.02 \pm 0.2$ | $30.23 \pm 1.7$ | $75.19 \pm 0.5$ | $42.35 \pm 0.3$ | $49.38 \pm 0.4$ | $85.88 \pm 0.1$ |
| AL | $\underline{14.48} \pm 0.7$ | $31.07 \pm 3.7$ | $\underline{76.21} \pm 0.2$ | $43.71 \pm 0.2$ | $50.79 \pm 0.2$ | $86.05 \pm 0.1$ |
| AVL | $14.01 \pm 0.3$ | $31.15 \pm 2.5$ | $75.84 \pm 0.9$ | $43.48 \pm 0.5$ | $50.59 \pm 0.8$ | $86.10 \pm 0.2$ |
| OL | $12.77 \pm 0.3$ | $30.87 \pm 1.1$ | $75.53 \pm 0.3$ | $44.03 \pm 0.3$ | $51.40 \pm 0.1$ | $86.03 \pm 0.1$ |
| OAL | $14.30 \pm 0.2$ | $30.79 \pm 1.5$ | $76.02 \pm 0.6$ | $44.13 \pm 0.5$ | $50.86 \pm 0.8$ | $86.09 \pm 0.2$ |
| MerlotR | $13.15 \pm 0.5$ | $\mathbf{32.53} \pm \mathbf{0.6}$ | $75.74 \pm 0.2$ | $44.52 \pm 0.8$ | $51.31 \pm 0.6$ | $86.22 \pm 0.2$ |
| ImageBind | $\mathbf{14.91} \pm \mathbf{0.2}$ | $\underline{31.31} \pm 3.1$ | $\mathbf{76.28} \pm \mathbf{0.5}$ | $34.15 \pm 0.5$ | $44.59 \pm 0.2$ | $84.11 \pm 0.1$ |
| MROH | – | – | – | $\mathbf{57.51}$ | $\mathbf{60.90}$ | $\mathbf{89.89}$ |

In Table 5 we present generalization performance on the validation split for next utterance prediction task and in Table 6 we demonstrate the generalization performance on validation split for atom classification task.

## E.2 GENERALIZATION PERFORMANCE OVER EPOCHS

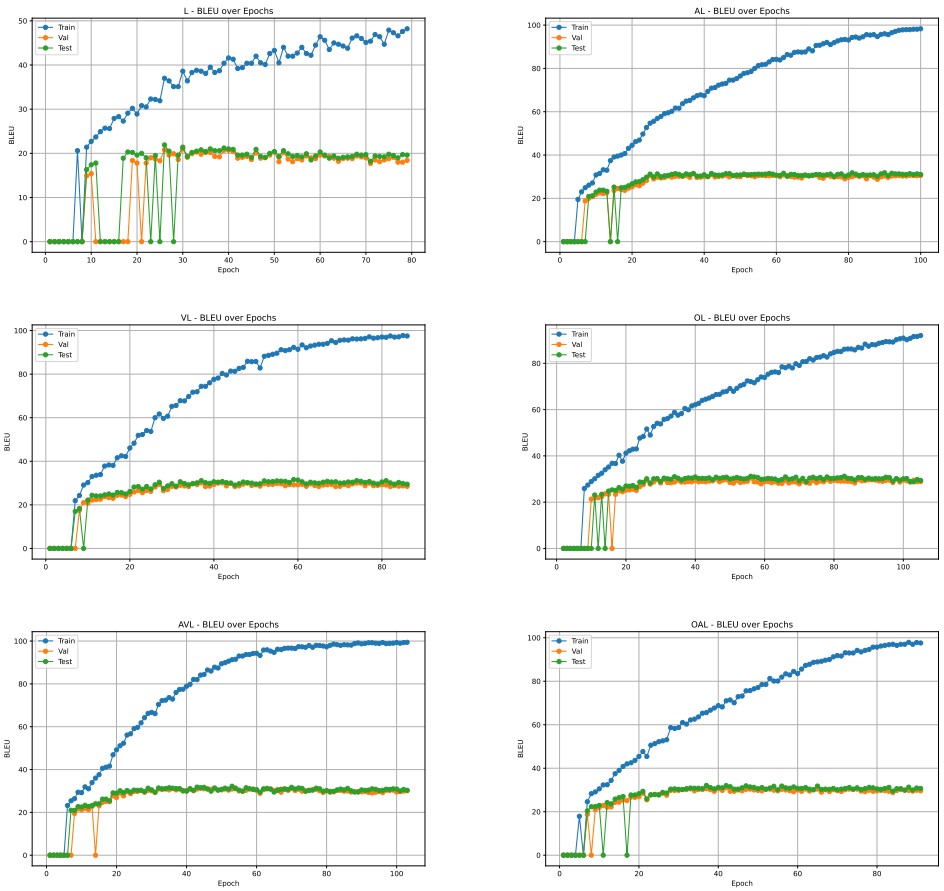

Figure 10: Generalization performance of the models over the epochs. Even though the training performance of a model improves on COMPACT, this does not necessarily mean that its validation and test performance will also become better due to the compositional nature of the COMPACT dataset.

In Fig.10, we report the BLEU scores of the models over the training, validation, and test splits at different epochs. These plots clearly demonstrate that in a compositional setup, models can perform well in the training set but this does not mean they can generalize to unseen distributions.

## E.3 PROMPTING ABLATIONS

### E.3.1 ADDITIONAL FEW-SHOT RESULTS

Table 7 and 8 offer interesting insights regarding few-shot compositional capabilities of IDEFICS and LLaMA2 models. First, we see significant performance discrepancy between IDEFICS and LLaMA2 on zero-shot prediction results. As IDEFICS additionally utilizes visual information over LLaMA2, it displays better zero-shot generalization capabilities. While LLaMA2 outperforms IDEFICS in one-shot and few-shot BLEU scores, contrastingly, IDEFICS outperforms LLaMA2 on BERT Scores. As LLaMA2 outperforms the LLM of IDEFICS (instruct-tuned LLaMA1) on many benchmarks (Touvron et al., 2023), we infer that LLaMA2 can imitate the vocabulary of few-shot examples better than IDEFICS, resulting in higher BLEU scores. However, higher BERT Scores imply that IDEFICS can reflect the semantics of the ground truth prediction better.

Table 7: Next utterance prediction results on test split for IDEFICS. As few-shot example count increases, performance improves on every metric consistently.

| $k$-shot | BLEU | EM | CA | BERT Score |
|--------|------|----|----|-----------|
| 0-shot | $8.98 \pm 0.2$ | $0.06 \pm 0.1$ | $0.12 \pm 0.0$ | $75.58 \pm 0.1$ |
| 1-shot | $20.25 \pm 0.2$ | $4.12 \pm 0.1$ | $5.30 \pm 0.1$ | $79.75 \pm 0.0$ |
| 3-shot | $24.85 \pm 0.7$ | $5.37 \pm 0.4$ | $7.20 \pm 0.3$ | $80.78 \pm 0.1$ |
| 5-shot | $25.64 \pm 0.4$ | $5.76 \pm 0.1$ | $7.89 \pm 0.5$ | $80.92 \pm 0.1$ |
| 8-shot | $26.18 \pm 0.3$ | $6.06 \pm 0.2$ | $7.92 \pm 0.3$ | $81.19 \pm 0.1$ |

Table 8: Next utterance prediction results on test split for LLaMA2. As few-shot example count increases, performance improves on every metric consistently.

| $k$-shot | BLEU | EM | CA | BERT Score |
|--------|------|----|----|-----------|
| 0-shot | $2.02 \pm 3.5$ | $0.13 \pm 0.1$ | $0.15 \pm 0.1$ | $71.68 \pm 0.1$ |
| 1-shot | $23.89 \pm 0.7$ | $3.98 \pm 0.2$ | $5.77 \pm 0.1$ | $77.90 \pm 0.2$ |
| 3-shot | $26.17 \pm 0.6$ | $5.07 \pm 0.1$ | $7.00 \pm 0.1$ | $78.35 \pm 0.1$ |
| 5-shot | $27.50 \pm 0.6$ | $5.36 \pm 0.6$ | $7.41 \pm 0.7$ | $78.76 \pm 0.2$ |
| 8-shot | $27.58 \pm 0.3$ | $5.60 \pm 0.2$ | $8.01 \pm 0.4$ | $78.95 \pm 0.1$ |

### E.3.2 FEW-SHOT EXAMPLE SELECTION

For few-shot example selection, rather than randomly picking $k$-shot examples, we employ a simple heuristic. As Liu et al. (2022) highlight that selecting similar examples improves in-context learning performance, we select the most similar $k$ examples as few-shot examples. The similarity measure between two examples is based on the noun and verb overlap. First, the intersection between the set of nouns and the set of verbs between the main example and all training examples is computed. If the sum of the cardinality of these sets is largest between the main example and a few-shot example, the few-shot example is the most similar example of the main example. We provide a validation comparison between the random example selection and our heuristic in Table 9.

Table 9: Next utterance prediction BLEU scores on validation split for IDEFICS for a single run. Greedy decoding is used and the best score is bolded.

| Strategy | 0-shot | 1-shot | 3-shot | 5-shot |
|----------|--------|--------|--------|--------|
| Random selection | **28.5** | 31.0 | 18.7 | 19.3 |
| Our heuristic | **28.5** | **34.8** | **23.4** | **23.5** |

### E.3.3 PROMPT TEMPLATE SELECTION

For IDEFICS, as images should be included in the prompt, the selection of prompt template is important. We compared two prompt templates (see Fig. 7 and Fig. 11) and after preliminary analysis, used the best performing template in our paper (see Table 10).

Table 10: Next utterance prediction results on validation split for IDEFICS. Overall, used template outperforms unused template.

| Template | BLEU | EM | CA | BERT Score |
|----------|------|----|----|-----------|
| Used template | $\mathbf{25.64} \pm \mathbf{0.4}$ | $5.76 \pm 0.1$ | $\mathbf{7.89} \pm \mathbf{0.5}$ | $\mathbf{80.92} \pm \mathbf{0.1}$ |
| Unused template | $22.19 \pm 0.3$ | $\mathbf{5.91} \pm \mathbf{0.4}$ | $7.14 \pm 0.5$ | $80.53 \pm 0.1$ |

```
Predict the next action narration given 3 sequential previous actions (image-narration
pairs) in a cooking video.
Narration 1: put down bowl Image 1: <Image 1>
Narration 2: move frying pan Image 2: <Image 2>
Narration 3: pick up spatula Image 3: <Image 3>
Narration 4: put down spatula

Narration 1: pick up tins Image 1: <Image 1>
Narration 2: put down tins Image 2: <Image 2>
Narration 3: move bowl Image 3: <Image 3>
Narration 4:
```

Figure 11: The unused alternative prompt template for IDEFICS evaluation.

### E.4 T-SNE VISUALIZATION FOR AUDIO AND VISUAL EMBEDDINGS

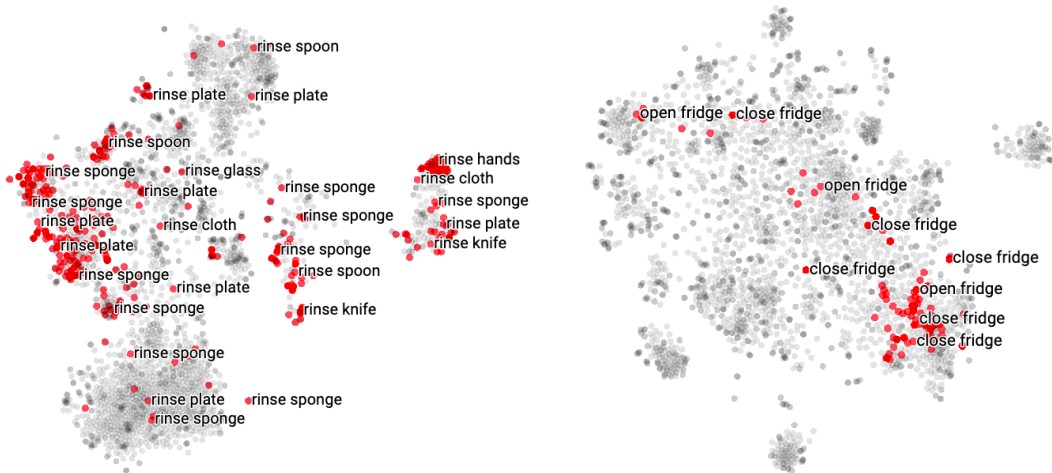

Figure 12: Feature projection to 2D space with t-SNE using raw audio and global visual features. On the left, audio space is shown with the verb *rinse* being specifically highlighted. On the right, visual space is given with noun FRIDGE being particularly highlighted. Sampled by most common compounds appearing at least 25 times in COMPACT, equally distributed for each compound ($N = 25$).

In order to understand the features we extracted via VGGSound and ResNet50 backbones and how well they encode the audio and visual spaces, we visualized the raw feature embeddings by projecting them to 2D space via t-SNE. In Figure 12, we highlight the compounds with the verb *rinse* and observe that audio features can meaningfully encode activities. Similarly, we analyzed the raw global visual embeddings and highlight the compounds with the noun FRIDGE, the visualization shows the extracted global visual embeddings can effectively encode visual surroundings.

## F    ETHICS STATEMENT

We curated our COMPACT dataset using the video clips from the published EPIC-Kitchens-100 dataset (Damen et al., 2022), which is publicly available. The videos that exist in this dataset were recorded voluntarily by the participants who were not financially rewarded.

