# OpenReview forum: "Compositional Generalization in Multimodal Foundation Models"
_ICLR.cc/2024/Conference — ICLR 2024 Conference Withdrawn Submission_

### Official Review · Reviewer_CZT6 · 2023-10-31

**Soundness:** 2 fair
**Presentation:** 2 fair
**Contribution:** 1 poor
**Rating:** 3
**Confidence:** 4

**Summary:**

The paper presents a study around the proficiency of multimodal foundation models in comprehending sequential activities. The authors curate the COMPACT dataset, which is a stratified subset of the Epic Kitchens dataset. They use this dataset to validate the performance of various models such as ImageBind, MERLOT Reserve, as well as Llama2.

**Strengths:**

The paper is clear and well-written. The study of multimodal models on natural video datasets is quite timely.
The idea of curating a subset of an existing dataset, rather than creating one from scratch, is a sound idea, especially since there are many similar datasets out there.

**Weaknesses:**

The paper is essentially an evaluation of well-defined tasks using an existing dataset and pre-trained models.
Out of the three tasks, the problem of action classification is only slightly different from next-utterance noun prediction and verb prediction (i.e., action is a combination of noun + verb, and the other two are predicted separately.)
Various pre-trained models are run in a zero-shot manner to predict the next entity in the sequence. While the evaluation results could be of interest to someone who is looking to build these models, the paper main content offers little beyond this evaluation.

**Questions:**

The result show that multimodality provides minimal advantage in the next noun and verb predictions tasks. However, the BLEU scores for the action classification are significantly improved. This is counterintuitive, since the actions are essentially nouns + verbs combinations. Therefore, if these individual units are incorrect, the overall BLEU score cannot be significantly better?

---

> ### Author Response · Authors · 2023-11-17
>
> Thank you for your thoughtful comments and for recognizing the timeliness and clarity of our study. We appreciate your insights and would like to address the points raised in your review.
>
> **Clarification on the CompAct Dataset:**
>
> As noted by other reviewers and reiterated in your review, we introduce the CompAct dataset, a dataset that is derived from the EpicKitchens-100 dataset. CompAct is not just a subset but reflects a targeted effort to explore compositional generalization in multimodal models.
>
> **Regarding the Action Classification Task:**
>
> We appreciate your insight into the Action Classification task. While this task might seem similar to Next-Utterance Prediction at first glance, its objectives are distinct. The Next-Utterance Prediction task measures the compositional generalization abilities of compounds expressed in a sentence, whereas the Action Classification task focuses on predicting isolated verbs and nouns. This distinction is crucial for understanding the models' prediction abilities and the influence of multimodality on compositional generalization, as we aim to dissect and understand the limitations of prior foundation models and the contribution of multimodality, as also highlighted by the reviewer w67G.
>
> **Clarification on Zero-Shot and Few-Shot Performances:**
>
> We have investigated not only the zero-shot performance of various pre-trained multimodal LLMs but also their few-shot (1-shot, 3-shot, 5-shot, and 8-shot) performances, as detailed in Appendix D2. This comprehensive analysis enhances our understanding of the models' capabilities across different learning scenarios.
>
> **Insight into BLEU Scores and Task Objectives:**
>
> Regarding your question about the BLEU scores in Action Classification, it's important to clarify that the objectives of the Action Classification and Next Utterance Prediction tasks are different. The Action Classification task focuses on individual atoms (verbs and nouns) in isolation, while the Next Utterance Prediction task is a language modeling task aimed at predicting novel sentences. We report Exact Match, Categorical Accuracy, and BertScore for the Action Classification task (Table 1) and BLEU scores for the Next Utterance Prediction task (Table 2). This differentiation in reporting metrics aligns with the distinct objectives of these tasks. Additionally, we are working on an analysis to make this distinction more clear and will share the results and update the paper after all additional analyses are completed.
>
> In conclusion, our work contributes a novel dataset and a unique approach to analyzing compositional generalization in multimodal models. We are grateful for your feedback, which has allowed us to clarify these aspects of our study. We believe our findings will be valuable to others in the field looking to build upon this research.

---

### Official Review · Reviewer_w67G · 2023-11-01

**Soundness:** 3 good
**Presentation:** 2 fair
**Contribution:** 2 fair
**Rating:** 5
**Confidence:** 5

**Summary:**

The paper aims to understand the capabilities of multimodal foundation models in terms of compositional generalization. To enable this study, the authors carefully constructed a dataset, CompAct (Compositional Activities), by reusing multimodal data and annotations from an existing benchmark, EpicKitchens-100, while defining new tasks and forming new training, validation, and testing splits.

Each data instance is an instructional video, containing video, audio, and step-by-step descriptions, where each step has a verb and a noun. CompAct is formed such that atomic concepts (verbs or nouns) are consistently distributed across the training and evaluation sets, while compositions of these atomic concepts are novel in the evaluation set.

CompAct allows for the diagnosis of the compositional generalization capabilities of both unimodal and multimodal models. The authors conduct an assessment of several unimodal and multimodal models, and their findings highlight the limited capabilities of prior foundation models for compositional generalization, as well as the importance of multi-modality over single-modality for certain challenging tasks.

**Strengths:**

1. The data distributions between training and evaluation in the CompAct benchmark are carefully controlled, allowing for the diagnosis of models' compositional generalization capabilities, which could be useful to the research community.

2. The authors present experimental results for approximately ten different unimodal or multimodal models. Some of the results are intriguing; for example, the language-only method outperforms the multimodal method in noun classification. However, for verb classification or next utterance prediction, the multimodal methods demonstrate superior performance.

3. The proposed method for curating train/eval splits to diagnose compositional generalization appears to be applicable to many other existing video datasets.

**Weaknesses:**

The authors have overlooked several works and benchmarks that are highly similar (see Questions below). Compared to these existing works, the contribution of this paper does not seem to be very significant. Additionally, the conclusions drawn from the experiments (e.g., recognition that compositional generalization is an area requiring improvement or that multi-modality could be more important than single-modality for certain challenging tasks) lack depth and insight.

**Questions:**

1. The CrossTask [1] dataset and its associated paper focus on an extremely similar study and settings. How does CompAct differ from the CrossTask benchmark? What unique contributions does your work make compared to the CrossTask paper?

2. The GAIN [2] benchmark is also a similar testing ground to the proposed CompAct in terms of evaluating models’ compositional generalizability and robustness under distribution shift. Unlike CompAct, whose atomic concepts are verbs or nouns and compositions are different verb-noun combinations, the atomic concepts in GAIN are steps, and the compositions are multi-step tasks. The authors should acknowledge these similar benchmarks and research efforts and clearly describe how this work advances the field.

3. Would experimental results on CompAct be translatable to these other similar benchmarks like GAIN, CrossTask, etc.? It would be interesting to find out.

4. Why are Maximum Compound Divergence and the Chernoff coefficient good measures for curating a dataset that requires compositional generalization, as opposed to other possible alternatives?

5. At the beginning of Section 4.1, it is mentioned that the first baseline is a text-only model to account for unexpected biases in CompAct. Why does a text-only model account for this?

6. There are many other instructional video datasets. Why was EpicKitchens-100 chosen?

7. For Noun Classification, there is the MROH baseline, which stands for Most Recent Object Heuristic. Why are there no results for the Most Recent Verb Heuristic in the Verb Classification task?

8. Why is the keyframe selection method different for ImageBind?


[1] Zhukov, Dimitri, Jean-Baptiste Alayrac, Ramazan Gokberk Cinbis, David Fouhey, Ivan Laptev, and Josef Sivic. "Cross-task weakly supervised learning from instructional videos." In Proceedings of the IEEE/CVF Conference on Computer Vision and Pattern Recognition, pp. 3537-3545. 2019.

[2] Li, Junlong, Guangyi Chen, Yansong Tang, Jinan Bao, Kun Zhang, Jie Zhou, and Jiwen Lu. "GAIN: On the Generalization of Instructional Action Understanding." In The Eleventh International Conference on Learning Representations. 2022.

---

> ### Author Response · Authors · 2023-11-17
>
> We appreciate your recognition of the value our work brings to the research community. We would like to respond to your questions and concerns about our paper.
>
> **Comparison with CrossTask and GAIN Benchmarks:**
>
> We acknowledge the similarities between our work and the CrossTask and GAIN benchmarks and appreciate your suggestions to distinguish our contributions more clearly. Unlike CrossTask, which focuses on cross-task generalization, our study centers on compositional generalization. We have updated the related work section to outline these differences and highlight the unique aspects of our approach. Similarly, while sharing similarities with GAIN in dataset formulation and the use of instructional videos, CompAct differs in the description of atomic concepts and the mathematical definition of out-of-distribution (OOD) scenarios. We have already outlined these distinctions in our paper and emphasized how CompAct specifically targets the evaluation of multimodal foundational models, furthering the understanding of compositional generalization.
>
> Exploring our models and approach on GAIN and CrossTask would indeed be an intriguing direction for future research. While we anticipate that our findings might generalize to GAIN due to similarities in data, the same may not hold for CrossTask, as it addresses a different aspect of generalization. We hope our work will inspire further investigations in these directions.
>
> **On Maximum Compound Divergence and Chernoff Coefficient:**
>
> Our choice of Maximum Compound Divergence (MCD) and the Chernoff coefficient follows the probabilistic and information-theoretic heuristic proposed by Keysers et al. [7]. These measures have proven effective in creating compositional splits while adhering to specific atomic and compositional constraints, e.g. in multilingual semantic parsing [8]. We are open to exploring other potential measures and welcome any suggestions you might have in this regard.
>
> **Rationale Behind Using Unimodal Models:**
>
> In line with previous research by Thomason et al. [9], as also pointed out in the paper in Section 4.1, we include unimodal models as a baseline to capture and reflect dataset biases. Unimodal models often perform surprisingly well, sometimes even outperforming multimodal counterparts, which can reflect systematic language-only biases in multimodal datasets.
>
> **Choice of EpicKitchens-100 Dataset:**
>
> The EpicKitchens-100 dataset was chosen due to its established reputation in the research community and its densely annotated instructions, offering a rich and diverse dataset. It also has a clear segmentation of instructions, including verb and noun annotations, making it an ideal candidate for curating the CompAct dataset, allowing us to leverage audio, vision, and text modalities effectively. The rationale behind choosing EpicKitchens-100 dataset is also discussed through the introduction section in the paper and detailed in Appendix A.
>
> **Inclusion of the Most Recent Verb Heuristic:**
>
> We appreciate your suggestion regarding the Most Recent Verb Heuristic baseline. Following your recommendation, we conducted an analysis with this baseline and will update our paper to include these findings after all additional analyses are completed. The results indicate a poor performance as shown below, which is somewhat expected as performing the same actions are less likely to occur consecutively in a sequence.
>
> > MRVH on CompAct test set:
>
> | EM | CA | BertScore |
> |---------|---------|-----------|
> | 2.39    | 9.61    | 73.60     |
>
>
> > MRVH on CompAct validation set:
>
> | EM | CA | BertScore |
> |---------|---------|-----------|
> | 2.56    | 9.66    | 73.45     |
>
>
> [7] Keysers, Daniel, Nathanael Schärli, Nathan Scales, Hylke Buisman, Daniel Furrer, Sergii Kashubin, Nikola Momchev et al. "Measuring Compositional Generalization: A Comprehensive Method on Realistic Data." In International Conference on Learning Representations. 2019.
>
> [8] Cui, Ruixiang, Rahul Aralikatte, Heather Lent, and Daniel Hershcovich. "Compositional generalization in multilingual semantic parsing over Wikidata." Transactions of the Association for Computational Linguistics 10 (2022): 937-955.
>
> [9] Thomason, Jesse, Daniel Gordon, and Yonatan Bisk. "Shifting the Baseline: Single Modality Performance on Visual Navigation & QA." In Proceedings of NAACL-HLT, pp. 1977-1983. 2019.

---

> > ### Comment · Reviewer_w67G · 2023-11-22
> >
> > I appreciate the authors' response and the experimental efforts spent. I would like to encourage the authors to report the Compound Divergence and Atom Divergence scores for the CrossTask and GAIN datasets in the revision. I shall retain my rating.

---

> > > ### Author Response · Authors · 2023-11-23
> > >
> > > Dear Reviewer,
> > >
> > > Thank you once again for your valuable feedback on our manuscript. In response to your remarks about GAIN and CrossTask benchmarks, we have conducted additional analyses on these datasets.
> > >
> > > We investigated whether CrossTask or GAIN could be considered for a compositional generalization benchmark. Lack of proper annotations for atoms and compounds and the number of instances seems to be a challenge to generate compositional splits for these benchmarks for instance, the initial analysis indicate that using a POS tagger for approximating atoms and generating compositional generalization splits for GAIN following a similar heuristic results with 10 instances in total for 3 window size in the input side and when this constraint is loosened to 2, only 60 instances can be generated for the GAIN dataset. We will include atom and compound divergence scores for GAIN and CrossTask but was not able to do so due to lack of text annotations.
> > >
> > > We sincerely hope that this addresses your concerns.

---

### Official Review · Reviewer_7kwf · 2023-11-01

**Soundness:** 3 good
**Presentation:** 3 good
**Contribution:** 3 good
**Rating:** 5
**Confidence:** 4

**Summary:**

This paper investigates the compositional generalization ability of multimodal approaches including baselines that are trained from scratch and existing large-scale pre-trained models. To do that, the introduces a new dataset called COMPACT, which is curated from the EK-100 dataset by ensuring that the individual concepts (verbs and nouns) exist across training and evaluation sets, while their compositions are novel in the evaluation set. The paper proposes two tasks for evaluation: (1) next utterance prediction: predicting the descriptions of the event in next video clip and (2) atom classification, predicting only the verb/noun involved in the event in the next video clip. The paper benchmarks several neural network models on the proposed tasks, including (train-from-scratch) text-only (unimodal) and multimodal models with different combinations of modalities as well as several large scale pretrained models using prompting techniques. Results show that all multimodal models surpass the text-only baseline.

**Strengths:**

The detailed strengths are as follows:
1. This paper is interesting because it is trying to understand the compositional generalization capabilities of foundation models. This is a crucial skill for intelligent agents and yet there are limited work and benchmarks proposed to investigate the question. Paper in this topic should be encouraged.
1. It investigates the important topic of compositional generalization capabilities in foundational models. This is a crucial skill for intelligent agents and yet there are limited research and benchmarks in this domain. Studies like this should be encouraged.
  - However, the paper appears to have limitations in addressing this issue for large-scale pre-trained foundational models. See weaknesses for details.
2. To answer this question, the paper presents a carefully curated novel dataset from real-world videos which could be much useful for future studies.
3. The paper also designs a set of multimodal models use different combinations of modalities (including unimodal) and different ways of fusing the multi-modal information. This investigation provides valuable insight on how multi-modality inputs could influence the performance of models' compositional generalization ability.

**Weaknesses:**

1. The paper does not sufficiently investigate the compositional generalization ability of **foundation** models. Addressing this is challenging due to the potential distributional discrepancies between training and testing splits during their pretraining, as noted in the paper. Consequently, emphasizing "foundation models" in the title may be somewhat overstated.
   - Could incorporating domain-specific fine-tuning offer additional insights?

2. The dataset's domain-specific nature results in text descriptions that lack diversity. As a result, unlike foundation LLM, language models trained on these specific tests might be prone to overfitting and lack reasoning skills. On the other hand, other modalities, such as the vision input processed by a pretrained ResNet model, inherently resist overfitting, potentially leading to enhanced generalization. Thus, the conclusion that multi-modality contributes to improvements and that visual features consistently enhance results could potentially be invalid.

**Questions:**

Please see weakness.

---

> ### Author Response · Authors · 2023-11-17
>
> Thank you for your detailed review and for recognizing the importance of our work in exploring the compositional generalization capabilities of foundation models. We value your feedback and would like to respond to your concerns.
>
> **On the Investigation of Compositional Generalization:**
>
> We agree that investigating the compositional generalization abilities of large-scale pre-trained foundational models is a hard task. However, despite strong performance of pretrained language models (LMs) across many tasks, they have been shown to struggle to compositional generalize (Furrer et al. [3]; Shaw et al. [4]; Bogin et al. [5]; Levy et al. [6]), when tested on their ability to process and generate novel combinations of previously observed elements. Our study, indeed, takes a step further and addresses the compositional generalization from a multimodal perspective by introducing the CompAct dataset.
>
> **Domain-Specific Fine-Tuning and Its Insights:**
>
> We agree that incorporating domain-specific fine-tuning could offer additional insights, particularly in enhancing model performance in specific contexts. While our study did not focus on domain-specific fine-tuning of multimodal large language models, we recognize its potential and believe that future work in this direction, using parameter-efficient tuning methods like LoRa, could yield better performance. However, the application of such techniques in encoder-only models remains an area for further research. We are currently running experiments on whether domain-specific fine-tuning could offer additional insights and will share our findings and update the paper after all additional analyses are completed.
>
> **Addressing Overfitting Concerns:**
>
> Regarding concerns about potential overfitting due to the dataset's domain-specific nature, we respectfully offer a different perspective. The train and test splits in our CompAct dataset are carefully curated to ensure diverse compound distributions, as illustrated in Figure 5. This deliberate design choice effectively mitigates overfitting risks across all models, including those utilizing additional modalities such as vision or audio. Furthermore, the pre-trained features used for these auxiliary modalities are integrated into our proposed models for further training (Figure 2), ensuring a robust evaluation of multimodal capabilities. The observed improvements over single-modal baselines across all multimodal models, as reported in Table 2, validate this approach. We are working on two additional analyses to address the overfitting concerns more clearly and will share our findings and update the paper after the additional analyses are completed.
>
> We appreciate the opportunity to clarify these aspects of our work and once again, thank you for your thoughtful and constructive feedback.
>
> [3] Furrer, Daniel, Marc van Zee, Nathan Scales, and Nathanael Schärli. "Compositional generalization in semantic parsing: Pre-training vs. specialized architectures." arXiv preprint arXiv:2007.08970 (2020).
>
> [4] Shaw, Peter, Ming-Wei Chang, Panupong Pasupat, and Kristina Toutanova. "Compositional Generalization and Natural Language Variation: Can a Semantic Parsing Approach Handle Both?." In Proceedings of the 59th Annual Meeting of the Association for Computational Linguistics and the 11th International Joint Conference on Natural Language Processing (Volume 1: Long Papers), pp. 922-938. 2021.
>
> [5] Bogin, Ben, Shivanshu Gupta, and Jonathan Berant. "Unobserved Local Structures Make Compositional Generalization Hard." In Proceedings of the 2022 Conference on Empirical Methods in Natural Language Processing, pp. 2731-2747. 2022.
>
> [6] Levy, Itay, Ben Bogin, and Jonathan Berant. "Diverse demonstrations improve in-context compositional generalization." arXiv preprint arXiv:2212.06800 (2022).

---

> > ### Author Response · Authors · 2023-11-23
> >
> > Dear Reviewer,
> >
> > Thank you once again for your valuable feedback on our manuscript. In response to your concerns, we have conducted additional analyses to further demonstrate our findings and methodology.
> >
> >
> > **On Domain-specific Fine-Tuning:**
> >
> > We conducted additional analysis whether domain-specific fine-tuning could offer additional insights and share our findings here:
> >
> > > Fine-tuning analysis on CompAct test set:
> >
> >
> > |                  | BLEU | EM | CA | BERT |
> > |------------------|-----------|---------|---------|-----------|
> > | IDEFICS 0-shot   |      8.98 |    0.06 |    0.12 |     75.58 |
> > | **IDEFICS LoRA** |     21.98 |    4.03 |    5.40 |     77.74 |
> > | LLAMA2 0-shot    |      2.02 |    0.13 |    0.15 |     71.68 |
> > | **LLAMA2 LoRA**  |      9.55 |    0.04 |    0.06 |     75.25 |
> >
> > For the domain fine-tuning tasks, we fine-tuned LLaMA 2 and IDEFICS models. We fine-tuned their Q and V attention weights with 4-bit QLoRA. As IDEFICS uses ViT, we fine-tune the image encoder as well. However, other than the tuned parameter size, every hyper-parameter is the same.
> >
> > The results show the same performance discrepancy as we previously reported. Both IDEFICS and LLAMA zero-shot results improve significantly compared to the previously reported zero-shot results.
> >
> > **Addressing Overfitting Concerns**
> >
> > We conducted another analysis to address the overfitting concerns. We replace the text encoder and text decoder with OPT-125M. Furthermore, in addition to the fusion parameters we also train cross-attention layers as in Ramos et al. CVPR 2023 [10]. Due to time constraints we experimented with VL and AVL baselines. We share our findings below:
> >
> > > Overfitting analysis on CompAct test set:
> >
> >
> > |            | BLEU | EM | CA | BERT |
> > |------------|-----------|---------|---------|-----------|
> > | VL_OPT     |      14.1 |    0.04 |    0.04 |     78.82 |
> > | AVL_OPT    |     14.57 |       0 |       0 |     78.83 |
> >
> > Based on the analysis we conduct, as can be seen from the results, even though we use a pretrained encoder and decoder, the performance does not improve suggesting that even using a pretrained encoder and decoder does not solve the generalization performance gap, therefore addressing the overfitting concerns.
> >
> > In Appendix D.2 of our revised manuscript, Figure 10 now illustrates the epoch-wise generalization performance of the models. This figure highlights an important aspect of our research: even as the training performance on the CompAct dataset improves, it does not necessarily correlate with enhanced validation and test performance. This is a direct consequence of the compositional nature of the CompAct dataset and further validates our approach to mitigating overfitting risks.
> >
> >
> > [10] - Ramos, Rita, Bruno Martins, Desmond Elliott, and Yova Kementchedjhieva. "SmallCap: lightweight image captioning prompted with retrieval augmentation." In Proceedings of the IEEE/CVF Conference on Computer Vision and Pattern Recognition, pp. 2840-2849. 2023.

---

### Official Review · Reviewer_JtuP · 2023-11-01

**Soundness:** 3 good
**Presentation:** 3 good
**Contribution:** 2 fair
**Rating:** 3
**Confidence:** 3

**Summary:**

This work studies the compositionality of vision language model. Specifically, it studied EPIC Kitchens-100 dataset and tailor the dataset with Maximum Compund Divergence heuristic for compositional generalization analysis.  The evaluations are performed on a number of methods, like VL, AL, OL, AVL, OAL, yet they all underperform the most recent object heuristic. The paper does not present novel algorithm or dataset, but provide an analysis for established ones.

**Strengths:**

The paper focus on studying the composition of foundation models on many variants, on Epic kitchen dataset that is tailored for composition evaluation.

**Weaknesses:**

The paper draws a conclusion that multimodal helps composition, yet from Table 1, the trend is not very clear.

**Questions:**

None.

---

> ### Author Response · Authors · 2023-11-17
>
> Thank you for your recognition of the focus of our work. We include responses to your concerns and include additional clarifications below.
>
> **Clarification on the CompAct Dataset:**
>
> This paper introduces the CompAct dataset, which is derived from the EK-100 dataset. CompAct is our contribution to evaluating the compositional generalization capabilities of multimodal foundation models. The dataset is not merely an application of existing methodologies but a targeted effort to understand model behaviors in compositional contexts, which expands the EK-100 dataset beyond its original purpose in a carefully controlled manner using maximum divergence splits.
>
> **Elaborating on the Results in Table 1:**
>
> We agree with you that none of the models outperform the simple MROH baseline on the object prediction task but we do not think this is a weakness of our work. We think this is an interesting finding that underscores the non-trivial nature of the task and highlights the challenges faced even by 9B parameter models. We believe that such insights are vital for advancing the field and understanding the limitations of existing models.
>
> Additionally, we want to emphasize that our aim with CompAct extends beyond individual atom prediction. The core of our paper includes the evaluation of models predicting the next entire utterances instead of individual atoms. We believe that the results in Table 2, contrary to Table 1, provide more compelling evidence that multimodality significantly enhances compositional generalization. More specifically, all of the multimodal baselines (O/A/VL) are 10 BLEU points better than the language-only baseline (L); and the multimodal foundation models MerlotR and ImageBind are 4–6 BLEU better than the LLaMA2 LLM.
>
> We hope this clarifies the main contributions of our paper.